# Interventions addressing family violence and mental illness or substance use in low- and middle-income countries: A systematic review

Jennifer J. Mootz[1,2] , Molly Fennig[3], Ali Giusto[1,2], Audrey Mumey[4], Claire M. Greene[5]  and Milton L. Wainberg[1,2]

[1]Department of Psychiatry, Columbia University, New York, USA; [2]New York State Psychiatric Institute, New York, USA; [3]Department of Psychiatry, Washington University in St. Louis, St. Louis, USA; [4]Department of Psychology, Columbia University, New York, USA and [5]Mailman School of Public Health, Columbia University, New York, USA

## Overview Review

**Keywords:**
domestic violence; child maltreatment; substance use; mental illness; intervention; low- and middle-income countries

**Corresponding author:**
Jennifer J. Mootz;
Email: Jennifer.mootz@nyspi.columbia.edu

## Abstract

Most family violence research has been conducted in high-income countries, although family violence rates are higher in low- and middle-income countries (LMICs), and outcomes more severe. Given the strong associations of family violence with substance use and mental illness, the aim of this systematic review was to examine interventions that targeted familial violence and at least one other condition of substance use or mental illness to determine effective treatments in LMICs. We conducted a systematic review of interventions that addressed family violence and mental illness or substance use. A committee of three researchers independently screened titles and abstracts and conducted full-text eligibility assessments. Two researchers conducted a risk of bias assessment. Data were extracted using a structured spreadsheet and narratively synthesized. Our search identified 29 articles produced from 19 studies conducted in 13 LMICs. Most ($n = 15$) studies randomized to study condition. Lack of blinding was the most common threat. The external validity of studies was generally poor. Fourteen studies had a primary intervention target of family violence, mental health, substance use, economic improvement, or HIV. None of the studies showed improvements in all intervention areas. Child maltreatment was less likely to be addressed than intimate partner violence (IPV). Targeted interventions for substance and mental health mostly improved primary outcomes, although they were less effective in reducing IPV. Evidence-based treatments must be rigorously evaluated before innovations in implementation can occur. Interventions overwhelmingly addressed IPV victimization and should consider how to work with couples and include men and children.

## Impact statement

Family violence affects between 15% and 71% of women and 75% of children worldwide and is a pervasive global health problem with a cascade of physical, psychological, and social consequences. While family violence has strong associations with substance use and mental illness, interventions that comprehensively address these related problems are lacking. Further, most family violence research has been conducted in high-income countries, although family violence rates are higher in low- and middle-income countries (LMICs), and outcomes are more severe. Despite the elevated need for LMICs, these settings have fewer resources to respond to family violence, and there is a paucity of trained mental health professionals in LMICs that contribute to large gaps in treatment. This systematic review of interventions that addressed family violence and mental illness or substance use in LMICs identified 29 articles from 19 studies conducted in 13 LMICs. The number of studies found shows that these co-occurring problems are being increasingly addressed in diverse global settings. Moreover, 15 of the studies were RCTs, demonstrating an overall low selection bias and an expansion in the rigorous study of these co-occurring problems. Despite these advances, several gaps remain. Most studies were underpowered and did not blind to conditions reducing the quality of the evidence. None of the studies showed improvements in all intervention areas. Child maltreatment was less likely to be addressed than IPV, and few studies targeted IPV perpetration. Interventions that targeted substance and mental health mostly improved primary outcomes, but they were less effective in reducing IPV. This review shows that much work is needed for the continued development and evaluation of integrated evidence-based treatments that address family violence and related substance use and mental health problems before innovations in implementation or scale-up efforts in LMICs can occur.

## Introduction

Violence is a pervasive global health problem with a cascade of physical, psychological, and social consequences accounting for 26 million disability-adjusted life years—years of life lost to illness, death, or disability (Kyu et al., 2018). Family violence consists of physical, sexual, or psychological abuse perpetrated by one person in the household against another (United Nations General Assembly, 1993; Asghar et al., 2017) and affects between 15% and 71% of women (Garcia-Moreno et al., 2006) and 75% of children worldwide (World Health Organization, 2019). Two common forms of family violence include intimate partner violence (IPV; violence between partners) and child maltreatment–violence or neglect perpetrated by adults against children (Krug et al., 2002).

Most family violence research has been conducted in high-income countries (HICs), although overall family violence rates are higher in low- and middle-income countries (LMICs) with over 90% of violence-related deaths occurring in LMICs (Matzopoulos et al., 2008). There are multiple drivers of family violence, such as inter-related inequitable gender norms pervasive in patriarchal contexts and poverty (Coll et al., 2020; Gilbert et al., 2022). Poverty is a risk factor for IPV perpetration and child maltreatment (Skeen and Tomlinson, 2013; Coll et al., 2020), and food insecurity increases odds of being exposed to family violence (Gibbs et al., 2018). Female death rates from violence increase as country income decreases (World Health Organization, 2019), and two-thirds of child deaths occur in LMICs (UNICEF, 2014). Additionally, consequences of family violence are more severe in low LMICs as compared to HICs (Garcia-Moreno et al., 2006; Sardinha et al., 2013). In LMICs, IPV is more likely to cause physical harm and involve sexual violence as compared to HICs (Garcia-Moreno et al., 2006). Thus, it may be especially important to examine interventions that reduce family violence in LMICs.

### Family violence, mental health conditions, and substance use

Given the multiple, intersecting levels in which family violence can occur, it is unsurprising that the relations among violence, mental illness, and substance misuse are dynamic with complex impacts on family systems. Exposure to family violence has been tied to mental illness, substance use, and risk for suicide (Dutton et al., 2017; Izaguirre and Calvete, 2018) across cultures (Vizcarra et al., 2010; Devries et al., 2011). IPV is one of the leading risk factors for common mental health conditions among women of reproductive age (Golding, 1999; Krug et al., 2002; Trevillion et al., 2012). IPV impacts child mental health and substance use outcomes via the direct effect of witnessing IPV and the indirect effect of poor maternal mental health (Herba et al., 2016; Stewart et al., 2016). Perpetration of IPV commonly co-occurs with violence against children. Child victims of violence are more likely to experience common mental conditions (depression, anxiety, and posttraumatic stress disorder [PTSD]) and substance use conditions as compared to non-exposed youth (Panyayong et al., 2018). Childhood abuse survivors or children who have witnessed IPV have been shown to be more likely to experience and perpetrate IPV themselves, placing their offspring at higher risk (Kimber et al., 2018). Finally, harmful substance use is a consequence of, and risk factor for, IPV (Foran and O'Leary, 2008; McCloskey et al., 2016).

### Present study

Family violence has strong bidirectional associations with mental illness and substance use, and integrated intervention strategies are needed to sustainably reduce family violence and associated psychosocial sequelae. Yet, there is a lack of effectiveness research on integrated interventions that simultaneously address these problems. Despite the elevated need for LMICs, these settings have fewer resources to respond to family violence and related mental health and substance use conditions. Protection, such as shelters, and violence prevention resources are scarce, and violence and mental health services are typically siloed, if available. Further, there is a paucity of trained mental health professionals in LMICs that contribute to large gaps in treatment (Saraceno et al., 2007; Bruckner et al., 2011). Given this reality, it is critical to consider how to treat co-occurring problems concurrently most efficiently and effectively. Therefore, this paper aims to systematically review interventions in LMICs that targeted familial violence and at least one other condition of harmful substance use or mental illness to determine effective treatments. To achieve this aim, we had two objectives: (1) describe the existing treatment evidence base and (2) identify gaps in the existing evidence.

## Method

### Inclusion and exclusion criteria

We registered the protocol (CRD42018085229) with PROSPERO in 2018. We included studies if they described any intervention examining at least one form of familial violence and at least one mental health and/or one substance use outcome among adults or families. Interventions were operationalized as including both behavioral health psychotherapies as well as interventions that have been shown to improve mental health outcomes irrespective of specific conditions (e.g., education or financial enhancement interventions; England et al., 2015). Next, interventions must have been implemented in an LMIC, as defined by the World Bank (2021). Finally, we included studies that had conducted a quantitative pre- and post-assessment of outcomes, at minimum. We excluded studies that evaluated interventions targeting only children under age 18 to review studies that focused on adult perpetration of violence against partners or children, used exclusively qualitative methods, and were not published in English.

### Search strategy and data extraction

Studies were identified by searching six electronic databases in 2018–2019 with no time parameters: Cochrane Central Register of Controlled Trials, EBSCO-SOC Index, EMBASE, Medline/Pubmed, Ovid/PsychINFO, and PILOTS. We also searched gray literature (Edlis, Mental Health Innovation Network, mhpss.net, OpenGrey, UNICEF, and WHO Global Index), reviewed the references of key reviews, and consulted with experts for recommendations.

Standardized search terms were applied in a stepped approach. Search syntax included terms and keywords related to the following: (a) each LMIC and setting type; (b) intervention; (c) mental health or substance use condition; and (d) family violence. English language filters were applied to searches. (See online Supplementary Table 1 for a full list of search terms.)

We used a multi-step process with Rayyan's online software database to compile and screen resulting titles and abstracts. First, three independent researchers reviewed the titles and abstracts for eligibility based on the predetermined criteria. Second, for the remaining articles, two independent researchers reviewed the full texts to determine eligibility. Discrepancies were resolved through

discussion and consensus following independent screening. Third, we extracted the relevant data into a spreadsheet developed to document setting, sample, methods, intervention, measures, outcomes, and limitations. Authors were contacted for any missing or incomplete data.

### Quality assessment

Study quality was evaluated using the Downs and Black (1998) and rated on four domains: Reporting (11 items), external validity (3 items), internal validity (14 items), and power (1 item). We reported on the following criteria: (1) clear reporting; (2) randomization; (3) allocation concealment; (4) blinding of participants and personnel; (5) blinding of outcome assessors; (6) appropriate analysis; (7) valid outcome measures; (8) adequate power; (9) external validity of the sample; and (10) external validity of intervention. Two independent reviewers assessed study quality. Discrepancies were resolved through discussion with a third reviewer.

### Results

Our search of six databases and gray literature resulted in 1514 non-duplicate articles. (See Figure 1 for the PRISMA flow chart). Out of those articles, 41 were identified as potentially relevant through the title and abstract screening and proceeded to full-text review. We identified four additional articles through citation tracking and contact with authors. The full-text screen of 45 articles resulted in 13 articles being excluded. Another three were excluded during data extraction. In the end, 29 articles from 19 studies in 13 LMICs were included.

### Characteristics of included studies

An overview of the included 19 studies can be found in Supplementary Table 2. Published between 2006 and 2019, studies were conducted in 13 countries across three continents (Africa = 11, Asia = 6, and South America = 2). Fifteen studies (79%) were randomized controlled trials. Two were pre/post without control and two were quasi-experimental.

### Participants

Four studies (21%) intervened with families, either through parent–child dyads (3, 16%) or couples (1, 5%). Three studies (16%) recruited adolescents (15 and over) and adults. The remaining 12 studies targeted female adults ($n = 7$, 37%), male adults (2, 11%), or both (3, 16%). Sample sizes ranged from 22 to 2,776. Counting families/dyads as one participant, 15 studies (79%) had over 100 participants.

### Quality assessment

All studies adequately reported details of the study objectives, procedures, and results (see Table 1). Regarding internal validity, 15 of the 19 included studies randomly allocated participants to the study conditions, reducing the risk of selection bias. However, only four studies reported whether the allocation process was concealed from participants and personnel. Lack of blinding, particularly of study participants and personnel, was the most common threat to the internal validity across studies. All studies appeared to apply appropriate statistical procedures and most reported using valid assessment tools to evaluate outcomes. Yet approximately half of the studies were underpowered or did not provide a power calculation. The external validity of the studies was generally poor. Apart from one study that screened all eligible women in the source population, participants who were recruited and/or those who were enrolled were not representative of the source population. Similarly, none of these studies evaluated interventions that are representative of services available to the source population, further limiting the external validity. We were unable to determine the external validity of the participants and/or interventions in 11 studies.

### Intervention description and outcomes

Most of the interventions ($n = 14$) had targeted primary outcomes aimed to reduce family violence, mental health, substance use, or improve participants' economic or HIV outcomes. These studies routinely measured reductions in symptoms on the other co-occurring problems. The remaining studies ($n = n = 5$) Ftested integrated interventions that explicitly addressed more than one outcome. See Supplementary Table 1 for a summary of interventions

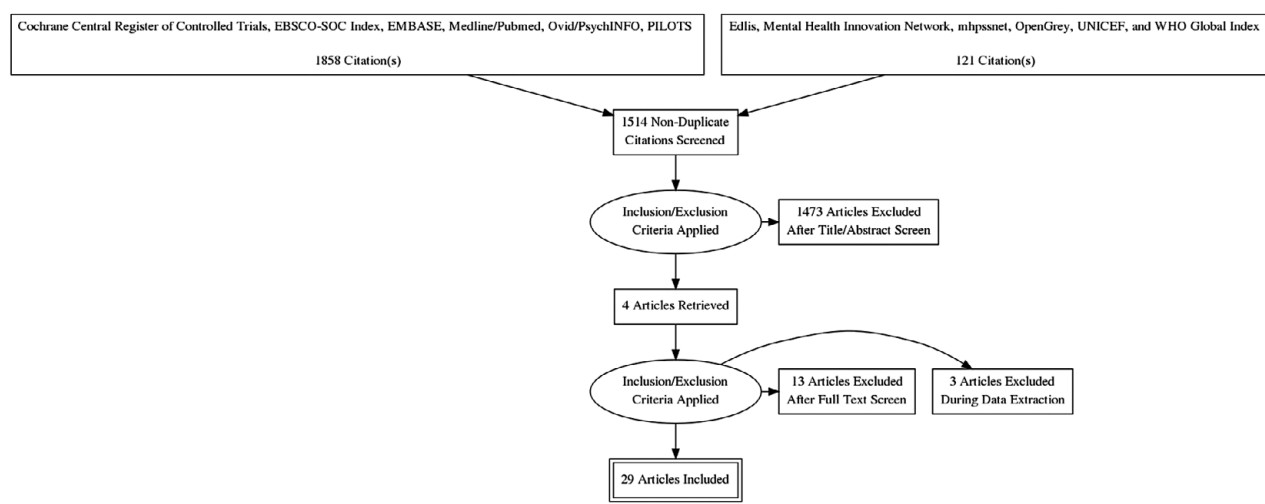

**Figure 1.** PRISMA flow chart

**Table 1.** Quality assessment

| Authors, Year | Clear Reporting | Randomized to Treatment | Allocation concealment | Blinding participants/ personnel | Blinding outcome assessment | Appropriate Analysis | Valid Outcome Measures | Adequate Power | External validity (sample) | External validity (intervention) |
|---|---|---|---|---|---|---|---|---|---|---|
| Meffert et al., 2014 | Y | Y | N | N | N | Y | Y | Un | N | N |
| Mutisya et al., 2017 | Y | N | N | Un | Un | Y | Y | Y | N | Un |
| Tiwari et al., 2010 | Y | Y | Y | Un | Y | Y | Y | Y | N | N |
| Gilbert et al., 2017 | Y | N | N | N | N | Y | Y | Un | N | Un |
| Jones et al., 2014 | Y | N | N | N | N | Y | N | Un | N | Un |
| Satyanarayana et al., 2016 | Y | Y | N | N | Y | Y | Y | Un | Un | N |
| Glass et al., 2017 | Y | Y | N | N | Un | Y | Y | Y | N | N |
| Lachman et al., 2017 | Y | Y | Y | N | Y | Y | Y | Y | N | Un |
| Witte et al., 2011 Carlson et al., 2012 | Y | Y | N | N | Y | Y | Y | Un | Un | Un |
| Tankard et al., 2019, | Y | Y | Un | N | Un | Y | N | Un | Un | N |
| Jewkes et al., 2014 | Y | N | N | N | N | Y | Y | N | Un | N |
| Patel et al., 2017 Weobong et al., 2017 | Y | Y | Y | Un | Y | Y | Y | Y | Un | Un |
| Nadkarni et al., 2017a Nadkarni et al., 2017b | Y | Y | Y | Y | Y | Y | Y | Y | N | Un |
| Jewkes et al., 2006 Jewkes et al., 2008 | Y | Y | N | N | N | Y | Y | Y | N | Un |
| Cluver et al., 2016 Cluver et al., 2018 | Y | Y | N | N | N | Y | Y | Y | N | N |
| Hidrobo et al., 2016 Buller et al., 2016 | Y | Y | Un | Un | Un | Y | Y | Un | N | N |
| Gupta et al., 2013 Annan et al., 2017 | Y | Y | N | N | N | Y | Y | N | N | N |
| Parcesepe, 2015 Parcesepe et al., 2016 | Y | Y | Un | Un | Un | Y | Y | Y | N | N |
| Chaudhury et al., 2016 Betancourt et al., 2017 | Y | Y | Un | Un | Y | Y | Y | Un | N | N |

*Note*: Green/Y=Yes; Red/N=No; Gray/Un = Unable to Determine.

(narratively described below), Table 2 for a summary of intervention outcomes of interest, and Table 3 for a description of intervention outcomes.

## Targeted family violence interventions

Three interventions focused on family violence. Most authors analyzed IPV items with a version of the Conflict Tactics Scale. The only study to focus primarily on reducing IPV and gender-based violence (GBV), Gilbert et al. (2017) implemented a two-session "Women Initiating New Goals of Safety" (WINGS) intervention with 66 women in Kyrgystan who reported using illicit drugs or binge drinking. WINGS is an SBIRT intervention (Screening, Brief Intervention, Referral to Treatment) that incorporates IPV and GBV screening, brief intervention, and referral to treatment. The authors conducted a within-group analysis of pre-/post assessments and observed changes in verbal and physical IPV, physical GBV, and linkage to IPV and GBV services. Reductions in sexual IPV and GBV did not occur, and verbal GBV worsened. They found significant improvements in illicit and injection drug use, but not binge drinking.

Two of the 19 studies targeted child maltreatment through parenting programs—Sinovuyo Teen (Cluver et al., 2016, 2018) and Sinovuyo Caring Families (Lachman et al., 2017)—delivered in South Africa. Both studies used standardized measures to analyze child maltreatment continuously post-treatment. Sinovuyo Teen, a 14-session program with 552 caregiver-adolescent dyads to prevent adolescent abuse, was compared to a one-day hygiene program in a cluster RCT. The program focused on improving the caregiver-adolescent relationship through time together, mutual praise, emotion management, problem-solving, and conflict management. They found reductions in abuse as reported on by youth and caregivers 1-month posttreatment and lower caregiver-reported abuse at 5- to 9-month follow-up. Five to nine months posttreatment they also showed improvements in parenting practices, caregiver depression, parenting stress, and social support, as well as caregiver and adolescent substance use. No significant changes were observed for neglect, adolescent mental health, or adolescent social support. Sinovuyo Caring Families, a 12-session, group-based program for reducing child maltreatment risk with low-income parents in Cape Town with children 3–8 years (n = 68 parent–child dyads), was compared to a waitlist control in an RCT. Sinovuyo Caring Families used behavioral parent management components to reduce maltreatment through improved parent–child relationships. No significant differences in child maltreatment were found but improvements in positive parenting were observed. Caregiver depression, parenting distress, and social support did not differ significantly across groups, nor were positive changes seen in observed and caregiver-reported child problem or positive behavior.

## Targeted substance use interventions

Two interventions had substance use as their primary target. L'Engle et al. (2014), Parcesepe (2015), and Parcesepe et al. (2016) conducted a multisite RCT to test a six-session alcohol harm reduction (one, 20-min session a month for 6 months) intervention against a control nutrition seminar with 818 adult female sex workers who were moderate-risk drinkers in Kenya. Facilitators were nurse counselors who received training in Motivational Interviewing. Improved reductions in alcohol use consumption frequency and in binge drinking episodes were evident in the intervention group at both the 6-month and 12-month follow-up timepoints. The intervention also showed stronger reductions in verbal, physical, and sexual abuse and being robbed by clients paying for sex work services. However, there were no differences between the intervention and control in improving physical and sexual violence with non-paying (i.e., intimate) partners.

In India, Nadkarni et al. (2017a) and Nadkarni et al. (2017b) examined outcomes between Counseling for Alcohol Problems (CAP), an intervention that uses a Motivational Interviewing approach and cognitive and behavioral skills for relapse prevention, and enhanced usual care. Lay counselors facilitated CAP in three phases for four weekly 30- to 45-min sessions with 377 male adults who presented at primary healthcare centers and had harmful drinking behaviors. CAP outperformed enhanced usual care in alcohol use outcomes of remission of harmful alcohol use (AUDIT < 8) and abstinence from alcohol, although not in problems related to alcohol or drug use. No differences between the intervention and usual care manifested in physical IPV and having made a suicide attempt in the past month.

## Targeted mental health interventions

Three RCTs studied the effects of interventions targeted toward mental health, showing mixed results. Most studies measured depression using structured and standardized surveys. (Chaudhury et al., 2016) and (Betancourt et al., 2017) reported findings from an RCT that examined differences between the Family Strengthening Intervention compared to social work support (treatment as usual) for HIV-positive caregivers and school-aged children ages 7–17 (n = 82 families) in Rwanda. The intervention consisted of six modules over the course of 6 months to reduce alcohol use and IPV among HIV-affected families. The authors measured IPV victimization and perpetration and alcohol use among adult caregivers (Chaudhury et al., 2016) and mental health outcomes in children (Betancourt et al., 2017). Both analyses found significant reductions of caregiver exposure to and perpetration of IPV, caregiver alcohol use, and youth and adult depression at the three-month follow-up (but not post-intervention). No differences between conditions existed for parenting, family connectedness, conduct, or functional impairment.

Patel et al. (2017) and (Weobong et al., 2017) conducted an RCT in India testing the Healthy Activity Programme with enhanced usual care against enhanced usual care only to reduce moderately severe to severe depression in 495 adults in primary care settings. Those randomized to the Healthy Activity Programme received 30–40-min, weekly individual sessions over 6–8 weeks consisting of behavioral activation, problem-solving, and activation of social networks facilitated by lay counselors with secondary education. Those who participated in the Healthy Activity Programme saw improved reductions in symptoms and remission of depression. Three months after enrollment, participants who received the Healthy Activity Programme reported less female victimization of physical IPV than those receiving usual care. Differences between the intervention and usual care arm in physical IPV subsided at the 12-month assessment timepoint. There were no differences in psychological/emotional IPV, combined physical and psychological/emotional IPV, or male victimization of IPV.

In Egypt, Meffert et al. (2014) tested the ability of Interpersonal Psychotherapy with 22 adult Sudanese refugees to reduce PTSD symptoms when compared to refugees in a waitlist control condition through an RCT. Nonspecialized Sudanese community therapists facilitated the therapy in a group modality twice a week for 3 weeks, adapted from the more standard 12-week duration for

**Table 2.** Summary of study intervention outcomes of interest

| Authors, Year | Intimate partner violence | | | | Mental health | | | | | Substance use | |
|---|---|---|---|---|---|---|---|---|---|---|---|
| | Physical | Sexual | Psychological | Economic | CM | Depression | Anxiety | PTSD | Other | Alcohol | Drugs |
| Buller *et al.*, 2016; Hidrobo *et al.*, 2016 | | | | | | | | | | | |
| Chaudhury *et al.*, 2016; Betancourt *et al.*, 2017 | | | | | | | | | | | |
| Glass *et al.*, 2017 | | | | | | | | | | | |
| Gupta *et al.*, 2013; Annan *et al.*, 2017 | | | | | | | | | | | |
| Jewkes *et al.*, 2006, 2008 | | | | | | | | | | | |
| Jones *et al.*, 2014 (HIV) | | | | | | | | | | | |
| L'Engle *et al.*, 2014; Parcesepe, 2015; Parcesepe *et al.*, 2016 | | | | | | | | | | | |
| Meffert *et al.*, 2014 | Violence towards household | | | | | | | | | | |
| Mutisya *et al.*, 2017 | | | | | | | | | | | |
| Nadkarni *et al.*, 2017a,b | | | | | | | | | | | |
| Patel *et al.*, 2017; Weobong *et al.*, 2017 | | | | | | | | | | | |
| Satyanarayana *et al.*, 2016 | | | | | | | | | | | |
| Tankard *et al.*, 2019 | IPV not differentiated | | | | | | | | | | |
| Tiwari *et al.*, 2010 | | | | | | | | | | | |
| Gilbert *et al.*, 2017 | | | | | | | | | | | |
| Jewkes *et al.*, 2014 | | | | | | | | | | | |
| Witte *et al.*, 2011; Carlson *et al.*, 2012 | | | | | | | | | | | |
| Cluver *et al.*, 2016; Cluver *et al.*, 2018 | | | | | | | | | | | |
| Lachman *et al.*, 2017 | | | | | | | | | | | |

*Notes:* = primary outcome of interest; = measured secondarily; = not measured as intervention outcome; CM = Child Maltreatment.

feasibility in a public health care and community delivery setting. Compared to the waitlist control, refugees who received Interpersonal Psychotherapy exhibited greater reductions in PTSD and state anger. There were no differences between the intervention and waitlist conditions regarding depression symptom and violence towards the household, although both were improved in the intervention condition. Differences between conditions in trait anger also did not evidence.

### *Targeted economic interventions*

Four of the interventions aimed to improve economic outcomes. Buller *et al.* (2016) and Hidrobo *et al.* (2016) evaluated an economic intervention through the World Food Programme with a cluster RCT. The program allotted aid in the form of cash, food, or food vouchers equaling the amount of $40 to 1,226 adolescent girls and women in urban settings once monthly over the course of 6 months. They found reductions in participants' experience of intimate partners' controlling behaviors and physical and/or sexual violence, but no significant change in emotional violence. In addition to IPV, the authors measured internal locus of control and happiness but found no change in these mental health constructs.

In the Democratic Republic of Congo, Glass *et al.* (2017) conducted an RCT with 833 adolescents/adults over age 16 that studied the effect of Pigs for Peace, a livestock asset transfer intervention, on victimization (girls/women) and perpetration (boys/men) of IPV, PTSD, anxiety, and depression. While they found significant

reductions in anxiety and PTSD symptoms, they observed no strong changes in victimization or perpetration of IPV or depression symptoms.

Annan *et al.* (2017) and Gupta *et al.* (2013) studied an intervention with 934 women to improve economic development and gender equity and reduce IPV and PTSD symptoms with an RCT in rural Côte d'Ivoire. The two-armed study compared (1) a weekly Village Savings and Loans Association economic empowerment program (group of women who collectively contribute to a shared fund that individual members can borrow from with interest) only with (2) the same enhanced with an eight-session gender dialog group that additionally met biweekly over the course of 4 months. The gender dialog groups were conducted with women and their male partners to reduce gender inequalities in households. Groups were co-facilitated by mixed-gender field agents who were specialists in gender-based violence and economic empowerment. The intervention arm showed a better ability to reduce economic abuse. For couples who had high adherence to the intervention (>75% meeting attendance), the enhanced intervention arm showed significantly better reductions in physical IPV and PTSD symptoms. No differences were observed for sexual IPV.

Finally, Tankard (2016) conducted an RCT with 1,800 women that examined the effects of assisting partnered women in urban settings in Colombia by opening an incentivized savings account. The intervention consisted of three free health checkups that included serology and a family planning consultation in addition to the option of opening a savings account with a contribution of

| Authors, Year | Intervention category and population | Measure(s) used | Target |
|---|---|---|---|
| **Between-group analysis** | | | |
| Buller et al., 2016; Hidrobo et al., 2016 | Targeted: Economic stability<br>Colombian refugees and poor Ecuadorian women in Northern Ecuador | **Family violence (primary):** *WHO Violence Against Women Instrument* | ***Aid given as cash, food, food vouchers vs control***<br>Aid significantly improved victimization of controlling behaviors (PTE = -0.07) and physical and/or sexual violence (PTE = -0.06) *versus* the control. There was no difference between aid and control for victimization of emotional violence (PTE = -0.04). |
| | | **Mental health (secondary):** Internal locus of control, 3 items; Happiness, 2 items | No differences were found for locus of control and happiness. |
| Chaudhury et al., 2016; Betancourt et al., 2017 | Targeted: Mental Health<br>At least one HIV+ caregiver in rural Rwanda and at least one school-aged child; willing to discuss HIV status with children | **Family violence (secondary):** *Conflict Tactics Scale* 22-item | ***Family strengthening intervention for HIV vs treatment as usual***<br>No differences were found for victimization and perpetration of emotional, physical, and sexual IPV at post-intervention. At 3-month follow-up, the Family Strengthening Intervention showed significant improvements over treatment as usual (effect size not reported). |
| | | **Mental health (primary):** *Center for Epidemiological Studies Depression Scale for Children; locally developed Conduct Problems scale; WHO Disability Assessment Schedule for Children* | For depression, no differences were found for youth or adults at post-intervention; at 3-month follow-up, the intervention showed improvements over treatment as usual for depression among youth (event ratio = 0.781) and adults (event ratio = 0.840). No differences were found in child conduct or functional impairment for youth or adults at post-intervention or 3-month follow up. |
| | | **Substance use (SU) (secondary):** *Alcohol Use Disorders Identification Test* | No differences were found at post-intervention in caregiver alcohol use. The intervention showed significant improvements over treatment as usual at the 3-month follow-up (effect size not reported). |
| Glass et al., 2017 | Targeted: Economic stability<br>Village households in Eastern Democratic Republic of Congo | **Family violence (secondary):** IPV questionnaire | ***Pigs for peace livestock vs delayed control***<br>No differences were found for victimization among women or perpetration among men in physical, sexual, psychological IPV |
| | | **Mental health (primary):** *Harvard Trauma Questionnaire, past 7 days*; Hopkins Symptom Checklist, past month | The intervention showed significant improvements in PTSD (ES = 0.21) and anxiety (ES = 0.15) over the control. No differences were found for depression (ES = 0.11). |
| Gupta et al., 2013; Annan et al., 2017 | Targeted: Economic empowerment/gender inequity<br>Women with partners in Côte d'Ivoire | **Family violence (primary):** Items from the WHO Multi-country study on Women's health and domestic violence | ***Village savings and loans association (VSLA) economic empowerment and savings program only vs VSLA + gender dialog group (VSLA + GDG)***<br>No differences were found for victimization of physical and/or sexual IPV overall (OR = 0.92), in high adherence (both partners attended more than 75% of meetings; effect size not reported), or low adherence (both partners attended less than 75% of meetings; effect size not reported); in victimization of physical IPV overall (effect size not reported) and in the low adherence condition (aOR = 0.93); in victimization of sexual IPV overall (effect size not reported), in the high adherence condition (aOR = 0.54), and in the low adherence condition (aOR = 0.85).<br>The intervention (economic empowerment + gender dialog) showed significant improvements over economic empowerment only in the high adherence condition for victimization of physical IPV (aOR = 0.45), victimization of economic abuse overall and the high (OR = 0.47) and low (OR = 0.31) adherence conditions of economic abuse |
| | | **Mental health (secondary):** *Harvard Trauma Questionnaire – PTSD section* | In a sample of 1,110, the intervention (economic empowerment + gender dialog) showed significant improvements in probable PTSD overall (OR = 0.61) and the high adherence condition (OR = 0.40). No differences were found in the low adherence condition (effect size not reported). For women who reported experiencing IPV at baseline (N = 202), there were no differences between study arms (OR = 0.72). |

(*Continued*)

**Table 3.** (*Continued*)

| Authors, Year | Intervention category and population | Measure(s) used | Target |
|---|---|---|---|
| Jewkes *et al.*, 2006, 2008 | Targeted: HIV<br>Adolescents, men, and women (ages 15-26) in South Africa | **Family violence (secondary):** 9-question structured questionnaire about IPV | ***Stepping stones for sexual health vs control session on HIV and safer sex***<br>No differences were found for men's perpetration or women's victimization of physical or sexual IPV at the 12-month (aOR = 0.73; 0.87) or 24-month (aOR = 0.62; 1.14) follow-up, respectively. |
| | | **Mental health (secondary):** *Center for Epidemiologic Studies Depression Scale* | No differences were found between the intervention and control for men's or women's depressive symptoms at the 12-month (aOR = 1.14; 1.32) or 24-month aOR = 0.52; 0.76) follow-up, respectively. |
| | | **Substance use (secondary):** *Alcohol Use Disorders Identification Test;* Ever used marijuana, mandrax, injectable drugs, substances that were sniffed, or other substances, 1 item | The intervention performed better than the control for male problem drinking at the 12-month follow-up (aOR = 0.68). Differences were not found for male problem drinking at the 24-month follow-up (aOR = 1.10), female problem drinking at the 12-month (aOR = 0.94) or 24-month (aOR = 1.40) follow-up, male misuse of drugs at the 12-month (aOR = 1.07) or 24-month (aOR = 0.50) follow-up, or women reporting misuse of drugs for their male partners at the 12-month (aOR = 0.60) or 24-month (aOR = 1.20) follow-up. |
| Jones *et al.*, 2014 | Targeted: HIV<br>Couples in Zambia with at least one member HIV+ | **Family violence (secondary):** *Conflict Tactics Scale,* past month IPV | ***The partner project-HIV couples, community health center staff-led vs research-staff-led vs control***<br>No differences were found among conditions for violence at the 6-month follow-up or extreme violence at the 6-month or 12-month follow-up (effect sizes not reported). No differences were found between community staff-led and research staff led arms at the 12-month follow-up for violence. However, both community staff led and research staff led arms performed better than the control condition for violence at the 12-month follow-up (effect size not reported). |
| | | **Substance use (secondary):** 5-item questionnaire about alcohol, past week | No differences were found among conditions for binge drinking (>5 drinks) at the 6-month or 12-month follow-up (effect sizes not reported). |
| L'Engle *et al.*, 2014; Parcesepe, 2015; Parcesepe *et al.*, 2016 | Targeted: Substance Use<br>Female sex workers in Mombasa who were moderate-risk drinkers and negative for STIs except HIV | **Family violence (secondary):** Self-reported, yes/no for "experienced" in past 30 days, 6-months post-intervention | ***Alcohol harm reduction intervention vs control nutrition seminar***<br>No differences were found for victimization of physical or sexual violence by a non-paying partner at post-intervention (physical violence OR = 0.64), 6-month (physical OR = 0.57; sexual aOR = 0.92), or 12-month (sexual aOR = 0.76) follow-up. |
| | | **Substance use (primary):** *Alcohol Use Disorders Identification Test* | The intervention performed better than the control condition for alcohol use frequency and binge drinking at the 6-month (aOR = 0.20; 0.13) and 12-month (aOR = 0.25; 0.18) follow-up, respectively. |
| Meffert *et al.*, 2014 | Targeted: Mental Health<br>Sudanese Refugees in Cairo with PTSD symptoms | **Family violence (secondary):** *Conflict Tactics Scale (CTS)* | ***Interpersonal therapy (IPT) vs Waitlist control***<br>No differences were found between Interpersonal Therapy and the waitlist control in violence towards the household (ES = -0.84) |
| | | **Mental health (primary):** *Harvard Trauma Questionnaire (HTQ); Beck Depression Inventory II (BDI-II),* past 2 weeks; *State–Trait Anger Expression Inventory (STAXI)* | Interpersonal Therapy performed better than the control in reducing trauma and PTSD (ES IPT = -2.52) and state anger (ES IPT = -1.21). No differences were found in depression symptoms (ES IPT = -2.38) or trait anger (ES IPT = -1.43). |
| Mutisya *et al.*, 2017 | Integrated: Mental Health + IPV<br>Pregnant women in Kenya | **Family violence (primary):** *Composite Abuse Scale* | ***Psychosocial intervention vs antenatal treatment as usual***<br>The psychosocial intervention performed better than treatment as usual for victimization of total IPV ($\eta^2 = 0.196$), physical violence ($\eta^2 = 0.305$), severe combined violence ($\eta^2 = 0.046$), emotional violence ($\eta^2 = 0.078$), and harassment ($\eta^2 = 0.086$). |
| | | **Mental health (secondary):** *Edinburgh Postnatal Depression Scale,* past 7 days | The intervention performed better than treatment as usual for postnatal depression ($\eta2 = 0.500$). |

(*Continued*)

**Table 3.** (*Continued*)

| Authors, Year | Intervention category and population | Measure(s) used | Target |
|---|---|---|---|
| Nadkarni *et al.*, 2017a, Nadkarni *et al.*, 2017b | Targeted: Substance Use<br>Men with harmful drinking (AUDIT: 12-19) in India | **Family violence (secondary):** Single-item violence, past 3 months | ***Counseling for alcohol problems with enhanced usual care vs enhanced usual care***<br>No differences were found for perpetration of physical IPV at 3-month (aOR = 0.81) or 12-month (aOR = 0.63) follow-up. |
| | | **Mental health (secondary):** Suicide attempt, past month, yes/no | No differences were found for having made a suicide attempt at 3-month (aPR = 1.8) or 12-month (aPR = 1.31) follow-up. |
| | | **Substance use (primary):** *Alcohol Use Disorders Identification Test,* past 14 days at 3-mo follow-up; *Short Inventory of Problems (SIP)* | The intervention condition performed better for remission of alcohol use (AUDIT<8) and no alcohol consumption at 3-month (aPR = 1.50; 1.71) and 12-month (aOR = 3.00; 1.92) follow-up, respectively. No differences were found for alcohol consumption among drinkers or reporting alcohol/drug use-related problems at 3-month (count ratio = 1.08; 0.99) and 12-month (AMD = -0.03; -0.40) follow-up, respectively. |
| Patel *et al.*, 2017; Weobong *et al.*, 2017 | Targeted: Mental Health<br>Adults in primary care with moderately severe to severe depression in India | **Family violence (secondary):** 2-item IPV questionnaire | ***Enhanced usual care with healthy activity programme vs enhanced usual care***<br>The Healthy Activity Programme performed better than enhanced usual care for women's victimization of Physical IPV at the 3-month follow-up (PR = 0.53). No differences were found for the 12-month follow-up (PR = 0.60). No differences were found for women's victimization of psychological or emotional IPV at the 3-month (PR = 0.76) or 12-month (PR = 0.75) follow-up. No differences were found for men's victimization of physical, psychological, or emotional IPV at 3-month (PR = 0.00 physical IPV; 0.67 psychological/ emotional IPV) or 12-month follow-up (PR = 0.82). |
| | | **Mental health (primary):** *Patient health questionnaire (PHQ-9); Beck Depression Inventory-II* | The intervention performed better than enhanced usual care for depression remission and reduction of depression symptom mean at 3-month (aPR = 1.61; ES = 0.48) and 12-month (aPR = 1.36; ES = 0.23) follow-up, respectively. |
| Satyanarayana *et al.*, 2016 | Integrated: IPV + Substance Use<br>Alcohol dependent, inpatient men who screened positive for IPV perpetration in the past 6 months in South India | **Family violence (primary):** *Index of Spouse Abuse* | ***Integrated cognitive-behavioral intervention vs treatment as usual***<br>The intervention performed better than treatment as usual for Severity of physical and non-physical IPV victimization among female spouses at 1-month (ES = 0.17) and 3-month (ES = 0.24) follow-up. |
| | | **Mental health (secondary):** *Depression, Anxiety and Stress Scale (DASS-21),* female partner; *Strengths and Difficulties Questionnaire (SDQ),* female partner | The intervention performed better than treatment as usual for depression (ES = 0.17), anxiety (ES = 0.15), and stress (ES = 0.07) among female spouses at 1-month and 3-month follow-up. No differences were found for child emotional/behavioral problems at 1-month or 3-month follow-up (effect sizes not reported). |
| | | **Substance use (primary):** *Severity of Alcohol Dependence Questionnaire (SADQ),* male participant | No differences were found for male alcohol use at 1-month or 3-month follow-up (effect sizes not reported). |
| Tankard *et al.*, 2019 | Targeted: Economic stability<br>Colombian adult women with a male partner and an interest in opening a savings account | **Family violence (secondary):** IPV questionnaire based on WHO Violence Against Women study | ***Savings account plus health visits vs control of no savings***<br>No differences were found for victimization of IPV (effect size not reported). |
| | | **Mental health (secondary):** *Stress, 4 items; Depression, 4 items; Anxiety, 4 items* | No differences were found in perceived stress or anxiety. The intervention performed better for depression (effect sizes not reported). |
| Tiwari *et al.*, 2010 | Integrated: Mental Health + IPV<br>Chinese women with histories of IPV | **Family violence (secondary):** *Chinese Conflict Tactics Scale*[V] | ***Advocacy intervention vs Control***<br>The advocacy intervention performed better than the control condition in reducing psychological IPV (Est. Difference = -1.87). No differences were found for physical (Est. Difference = -0.35) or sexual (Est. Difference = -0.12) IPV. |
| | | **Mental health (primary):** *Chinese Beck Depression Inventory (C-BDI-II),* past 2 weeks; *Interpersonal Support Evaluation List (ISEL)* | The intervention performed better than the control condition in reducing depressive symptoms (Est. Difference = -2.66) and improving perceptions of social support (Est. Difference = 2.18). |

| Authors, Year | Intervention category and population | Measure(s) used | Target |
|---|---|---|---|
| **Within-group analysis** | | | |
| Gilbert et al., 2017 | Targeted: IPV<br>Women who used illicit drugs or binge drank in past 6 months in Kyrgyzstan | **Family violence (primary):** *Revised Conflict Tactics Scale (CTS-R)*, past 90 days; One-item receipt of IPV/GBV services, past 90 days | **WINGS: Women initiating new goals of safety**<br>Wings showed significant improvements in victimization of verbal (aIRR = 0.79) and physical (aIRR = 0.41) IPV, verbal (aIRR = 1.34) and physical (aIRR = 0.73) GBV, and linkage to IPV/GBV services (aOR = 12.3). No significant improvements found for victimization of sexual IPV (aIRR = 0.94) or GBV (aIRR = 0.89). |
| | | **Substance use (secondary):** *Risk Behavior Assessment,* past 90 days | WINGS showed significant reductions in any illicit drug use (aOR = 0.35) and any injection drug use (aOR = 0.39) in the past 30 days. No significant improvements found for reduction of binge drinking (aOR = 0.50) in the past 90 days. |
| Jewkes et al., 2014 | Integrated: Economic Empowerment + HIV + IPV<br>Young men and women in urban informal settlements in South Africa | **Family violence (primary):** IPV Self-report questionnaire | ***Stepping stones + Creating futures***<br>Significant improvements found for reductions in women's victimization of sexual IPV. No significant reductions found for men's perpetration of physical or sexual IPV or women's victimization of physical IPV (effect sizes not reported). |
| | | **Mental health (primary):** *Center for Epidemiological Studies Depression Scale*; Suicidal thoughts | Significant reductions found for men's depressive symptoms and suicidal thoughts in the past four weeks. No significant improvements found for women's depressive symptoms or suicidal thoughts (effect sizes not reported). |
| | | **Substance use (secondary):** Adapted *Alcohol Use Disorder Identification test* | Women's alcohol use significantly worsened. No changes were found in men's alcohol use (effect sizes not reported). |
| Witte et al., 2011;<br>Carlson et al., 2012 | Integrated: IPV + Substance Use<br>Female sex workers in Mongolia | **Family violence (primary):** *Revised Conflict Tactics Scale*, past 90 days, paying and non-paying partners | ***HIV sexual risk reduction vs HIV-sexual risk reduction + motivational interviewing vs control***<br>Results reported for individual study arms. Significant improvements in victimization of physical (HIV risk reduction + motivational interviewing OR = 0.29; HIV risk reduction OR = 0.11; control OR = 0.15), sexual (HIV risk reduction + motivational interviewing OR = 0.15; HIV risk reduction OR = 0.06; control OR = .05), and physical or sexual (HIV risk reduction + motivational interviewing OR = 0.20; HIV risk reduction OR = 0.14; control OR = 0.46) IPV were found in all three arms. |
| | | **Substance use (primary):** *Alcohol Use Disorders Identification Test* | Significant improvements found for harmful drinking in all three study arms (HIV risk reduction + motivational interviewing RRR = 0.60; HIV risk reduction RRR = 0.60; control RRR = 0.54). |
| **Child maltreatment studies** | | | |
| Cluver et al., 2016;<br>Cluver et al., 2018 | Targeted: Child Maltreatment<br>Adolescents (10-18 years) and caregivers in South Africa | **Family violence (primary):** *International Society for Prevention of Child Abuse and Neglect Screening Tool for Trials (ICAST-Trial)* & *Alabama Parenting Questionnaire* | ***Sinovuyu teen programme (STP) vs. Hygiene control (C)***<br>The Sinovuyu Teen Programme performed better than the control condition in reducing physical and emotional abuse at 1-month as reported by adults (IRR = 0.39) and youth (IRR = 0.71) and at the 5–9 month follow-up, as reported by adults (IRR = 0.55). No differences were found between conditions at the 5-9 month follow-up as reported by youth (IRR = 0.90). Also, no differences were found for neglect, as reported by adults (IRR = 0.31) and youth (IRR = 1.46). The intervention performed better than the control condition in reducing corporal punishment, as reported by adults (IRR = 0.55). However, according to youth reporting, no differences in corporal punishment were evident (IRR = 1.05). |
| | | **Mental health (secondary):** *Child Behavior Checklist*, rule-breaking and aggression subscales; *Parental Stress Scale; Children's Depression Inventory/Mini International Neuropsychiatric Interview-Kid; Centre for* | No differences between conditions were found for adolescent externalizing behavior, as reported by youth (d = 0.12) and adults (d = -0.16), and youth-reported depression/suicidality (IRR = 1.02), or youth-reported increased social support at 5-9 month follow-up (d = -0.04). The intervention performed better than the control among caregivers for reducing parenting stress (d = -0.37), depression, (d = -0.33), and increasing social support (d = 0.31). |

**Table 3.** (Continued)

| Authors, Year | Intervention category and population | Measure(s) used | Target |
|---|---|---|---|
| | | *Epidemiologic Studies Depression Scale; Medical Outcomes Study Social Support Survey* | |
| | | **Substance use (secondary):** *Alcohol Use Disorder Identification test; WHO Global School-based Health Survey* | The intervention performed better than the control in reducing alcohol and drug use among adults (IRR = 0.67) and youth (IRR = 0.55). |
| Lachman et al., 2017 | Targeted: Child Maltreatment<br>Low-income parent–child (3-8 years) dyads in South Africa | **Family violence (primary):** *Parent–Child Conflict Tactics Scale; Parenting Young Children Scale; Sinovuyo Observational Coding System* | ***Sinovuyo caring families program for young children (SCFP) vs. waitlist control (C)***<br>The intervention performed better than the waitlist control in self-report of positive parenting (dX = 0.63). No differences were found between conditions for self-report of harsh parenting (dX = 0.02) and observed negative (dX = –0.34) and positive (dX = 0.13) parenting. |
| | | **Mental health (secondary):** *Parenting Stress Index-Short From; Beck Depression Inventory-II; Eyberg Child Behavior Inventory; Sinovuyo Observational Coding System; Multidimensional Scale of Perceived Social Support (MSPSS)* | No differences between conditions were found in parenting stress (dX = –0.23), parental depression (dX = 0.12), child problem behavior (dX = –0.11), observed child negative behavior (dX = 0.30), and perceived parental social support (dX = 0.27). The intervention performed worse than the control in observed child positive behavior (dX = –0.56). |

**Intervention Category:** Integrated = treatment focused on reduction of violence and mental health or substance use; Targeted = focused intervention to address one outcome.
**Measure(s) used:** primary = measured as a primary outcome of interest; secondary = measured as a secondary outcome of interest.
**Target:** AMD = adjusted means difference; aPR = adjusted prevalence ratio; OR = odds ratio; aOR = adjusted odds ratio; IRR = Incidence rate ratio; eOR = estimated odds ratio; RCS = residualized change score; dX = bias correct Cohen's d; PTE = pooled treatment effect; ES = effect size not specified; NR = Not reported.

10,000 pesos ($5) and matched savings. The control condition included health checkups without the savings account option. Assessments were given at pre-intervention and nine- and 18-months post-baseline. The only difference between conditions occurred with the reduction of depression symptoms. Other mental health constructs and IPV victimization showed no differences between conditions.

### Targeted HIV interventions

Two interventions targeted HIV prevention and response. Jewkes et al. (2006, 2008) conducted a cluster RCT with 232 young adults in South Africa, comparing the intervention Stepping Stones to a control group session. Stepping Stones was comprised of 17 three-hour group sessions conducted over the course of 3–12 weeks. The intervention aimed to improve sexual health with a focus on gender-equitable relationships and communication through education about sexual health, sexually transmitted infections, gender-based violence, and relationships. They found that the intervention showed stronger improvements in male problem drinking at the 12-month follow-up timepoint. Other areas—physical and sexual IPV, depression, female problem drinking and drug misuse—showed no differences between the intervention and control.

Jones et al. (2014) facilitated a couple-based intervention called The Partner Project with 394 couples in Zambia where at least one partner had HIV. They compared intervention arms to see if there were differences according to the specialization level of facilitator. Groups led by research staff were compared to a second arm facilitated by nonspecialized community workers and a third arm that served as a control. The intervention consisted of four gender-specific group sessions that provided focused on sexual risk prevention through training and psychoeducation on negotiation and communication strategies. They found that both the researcher and community-led intervention arms outperformed the control arm in reductions of past-month IPV at the 12-month follow-up assessment. They observed no differences in extreme violence or binge drinking.

### Integrated family violence, mental health, and substance use interventions and outcomes

Two interventions targeted both family violence and mental health. Both interventions reduced mental health symptoms and one improved IPV. In Kenya, Mutisya et al. (2017) conducted a quasi-experimental study with 283 pregnant women in their first and second trimester who sought antenatal services in primary healthcare settings and reported experiencing any form of IPV. The psychosocial intervention consisted of psychoeducation about IPV and its effects on the pregnancy, empathic listening, and referrals for gender-based violence resources. The intervention was delivered in three 30-min sessions over the course of 5 months. Its comparison, treatment as usual, consisted of antenatal services with a list of GBV services. Benefits of the psychosocial intervention evidenced in all forms of IPV, although total and physical IPV showed the strongest effect sizes. The psychosocial intervention also demonstrated better improvements in severe combined IPV, emotional violence, and harassment than usual care. However, the effect sizes were negligible. The intervention also showed significantly stronger reductions in depressive symptoms.

In China, Tiwari et al. (2010) conducted an RCT with 200 women who had histories of IPV. The advocacy intervention consisted of empowerment (decision-making, problem-solving,

and protection) and telephone social support. To improve decision-making, facilitators provided information on cycles of violence, legal matters, and community resources so women can better recognize escalating violence to enact safety planning. The empowerment component was one 30-min session and social support entailed 12 weekly calls from a research assistant. Intervention recipients had access to the community center's childcare, health care, and recreational services. Women in the control group received community center services without specialized violence services. The advocacy intervention had stronger effects in reducing depressive symptoms and psychological IPV and increasing social support. Differences in physical or sexual IPV did not emerge between conditions.

Two studies examined interventions that targeted IPV and alcohol use. Both interventions reduced IPV. However, reductions in alcohol use were mixed. Satyanarayana *et al.* (2016) conducted an RCT in South India to test an Integrated Cognitive Behavioral Intervention (ICBI) to address IPV perpetration and alcohol use among 177 male inpatients diagnosed with Alcohol Dependence Syndrome, had a child under age 16, and had perpetrated some form of IPV in the past 6 months. The facilitators had graduate degrees in clinical psychology and were certified in ICBI. Treatment as usual included psychopharmacology and one session of psychoeducation on alcohol dependence symptoms and treatment options. ICBI encompassed eight 45–60-min sessions that addressed the relation between IPV and alcohol and identification of triggers, consequences, and prevention of IPV and alcohol. Men learned cognitive restructuring, anger management, and relaxation. There were small to medium effects of ICBI in reducing severity of IPV and depression, stress, and anxiety among the participants' wives. Yet, no differences emerged between conditions in male alcohol use and child emotional and behavioral problems.

Witte *et al.* (2011) and Carlson *et al.* (2012) facilitated an RCT that tested an intervention aimed to reduce sexual risk behaviors, harmful drinking, and IPV victimization among 166 female sex workers in Mongolia. The two active treatment arms were relationship-focused and consisted of (1) four weekly group sessions that promoted knowledge and skills building for HIV and sexually transmitted illness risk reduction and (2) the first arm enhanced with two additional sessions of Motivational Interviewing. These treatments were compared to a third control wellness promotion group. All conditions effectively reduced harmful alcohol consumption (Witte *et al.*, 2011) and physical and sexual IPV (Carlson *et al.*, 2012). However, no differences were observed between the intervention arms and the control condition.

Finally, Jewkes *et al.* (2014) targeted economic empowerment, HIV, and IPV by combining Stepping Stones (described under targeted HIV interventions) and Creating Futures, a same-sex group livelihoods intervention. Their quasi-experimental interrupted time series study occurred in two urban informal settlements in South Africa with 232 young adults aged 18 to 34. Stepping Stones was adapted to focus on HIV and violence prevention and delivered in 10 three-hour group sessions to improve communication and develop gender-equitable relationships. Creating Futures was conducted over 11 three-hour sessions and included participatory learning activities to critically reflect on current resources and how to build them. No reductions were observed in men's perpetration of physical or sexual IPV, although women reported less sexual IPV victimization. Men showed improvements in depressive symptoms and suicidal ideation. However, women did not report similar gains in mental health, although they showed

reductions in alcohol use. Men's alcohol use, in contrast, did not significantly reduce.

## Discussion

The objective of this systematic review was to examine interventions that targeted familial violence and at least one other condition of harmful substance use or mental illness to determine effective treatments in LMICs by describing the evidence base and identifying gaps. Nineteen studies (29 articles) out of a potential 1,514 articles from 13 LMICs met the inclusion criteria. A recent review of mental health interventions' effects on IPV found seven studies all from middle-income countries and noted lack of representativeness of low-income countries as a critical gap (Tol *et al.*, 2019). Thus, it appears that these co-occurring problems are being increasingly addressed in diverse settings. Moreover, 15 of the studies were RCTs, demonstrating an overall low selection bias and an expansion in the rigorous study of these co-occurring problems in LMICs.

The most substantial concerns regarding the quality of the studies were that many studies were underpowered or did not report a power calculation, lacked blinding to condition, and were not representative of the source population. In behavioral interventions, having a placebo control may not be ethical and a placebo may be more difficult to replicate with more complex behavioral interventions (Friedberg *et al.*, 2010; Monaghan *et al.*, 2021). Likely for this reason, most of the studies in this review had usual care control arms in which blinding participants could be particularly difficult. Providers of behavioral interventions oftentimes cannot be blinded when the outcome of interest, such as symptom change, informs the course of the intervention. Blinding of outcome assessors is possibly the most critical for curbing bias since expectations can influence evaluation and inflate effect sizes. A review of 252 trials found that out of the 125 trials that assessed outcomes, only 26% reported using blinding. The use of blinding was largely dependent on use of an independent assessor (Kahan *et al.*, 2015). All these challenges are augmented in LMICs where of a lack of human resources and weak behavioral health infrastructures may be a local reality. Funding may not cover the costs of hiring an independent assessor. Office spaces are often shared, and to streamline human resources, study team meetings might consist of many team members, including research assistants who are involved in several aspects of the study, such as assessing eligibility criteria, performing outcome assessments, and randomizing participants to study arms. Where possible, it is recommended to hire an independent assessor to conduct outcome evaluations and avoid sharing study hypotheses and specific outcomes of interest, including in online study descriptions (Friedberg *et al.*, 2010).

Interventions and measurement of outcomes varied widely. Interventions differed in duration (three weeks to two years), dose, and modality. Most studies (*n* = 15) had a singular intervention target. Still, there was no consistent trend of effectiveness across interventions. While implementation of evidence-based treatments with standardized follow-up timepoints and treatment durations could facilitate comparison across studies in future research, a common assertion in global mental health scale-up efforts (Bemme and D'souza, 2014), there are numerous challenges to standardization. The first issue is relevance and acceptability, as local cultural contexts shape how mental health conditions are conceptualized and expressed symptomatically (Mendenhall *et al.*, 2016). In this review of studies, most interventions were

theoretically informed and locally developed or adapted, which may explain the heterogeneity of outcomes regarding effectiveness. Other challenges to standardization of measurement across LMICs are that adopting and implementing new measures is costly and establishing expert consensus is difficult (Liao and Quintana, 2021). None of the studies showed improvements in all three areas of family violence, mental health, or substance use. Future intervention development should focus on strengthening theoretical explanations for how family violence, mental health, and substance use relate to one another and identifying core mechanisms of therapeutic action. While several authors assessed outcomes at multiple time points and follow-up, efforts were placed on understanding effect sizes rather than mediational effects. Use of path analyses in future studies could help untangle these associations further. Moreover, conducting meta-analyses to synthesize effectiveness and look more closely at factors such as intervention length and dose of sessions will be an important next step.

Outcomes for family violence also varied with few interventions improving all family violence domains. Many studies measured IPV perpetration. However, it was often unclear how interventions specifically were adapted to, or targeted, men's perpetration of IPV. Only one study (Patel *et al.*, 2017; Weobong *et al.*, 2017) assessed IPV victimization among men. Globally, there is a lack of evidence-based treatments shown to be effective for reducing IPV perpetration. Research in high-income settings has shown that situational IPV—discord and conflict in couples that escalates to mild or moderate physical violence—is often bidirectional (Langhinrichsen-Rohling *et al.*, 2012). Our research in Uganda has shown that approximately 35% of women reported perpetrating IPV towards their male partners and that perpetration was associated with victimization (authors blinded). Couple-based interventions to reduce conflict have been shown to be effective (Stith and McCollum, 2011) and are needed. Only one study (Jones *et al.*, 2014) in this review used a couple-based intervention to improve HIV outcomes. Interventions that work with the couple system are needed and may hold higher cultural relevance, given emphasis on family unity and stigma about separation (authors blinded). While many researchers employed standardized measures to assess violence, if researchers elected not to use a full standardized instrument, the rationale for selecting certain IPV items was sometimes unclear.

Child maltreatment was less likely to be addressed than IPV. Two studies (Cluver *et al.*, 2016; Lachman *et al.*, 2017) in South Africa specifically aimed at reducing child maltreatment. Only one study (Meffert *et al.*, 2014) more broadly measured violence towards the household, although there is a known association between perpetration of IPV and child maltreatment (Namy *et al.*, 2017; Mootz *et al.*, 2019). Based on the available data presented in this review, it is difficult to make conclusive takeaways about how family violence and mental health or substance use program impact youth mental health. While the developmental literature suggests these factors—violence, maltreatment, and caregiver mental health conditions—influence the development of youth mental health conditions, the studies on IPV did not often include measures of child mental health. It is possible that if they did, improvements in child mental health would be seen. Qualitative research in Kenya has shown that intervening to reduce fathers' alcohol use provided mental health benefits to their families (Giusto *et al.*, 2021). Future research should examine downstream effects of reducing IPV and adult mental health and substance use on children. On the other hand, child maltreatment studies did include measures of youth mental health yet results on mental health were mixed even with observed changes in caregiver violence. It is

possible that a longer follow-up period is needed to capture potential improvements in children's mental health. In other words, changes in the family system may take longer to influence youth mental health. Consideration of whether family-level interventions outperform couple or individual modalities could help mobilize the few resources available for mental health in low-resource settings.

Several studies in this review targeted substance use or mental health. The substance use interventions (L'Engle *et al.*, 2014; Parcesepe *et al.*, 2016; Nadkarni *et al.*, 2017b) outperformed control conditions in reductions in alcohol use. However, no differences between conditions in IPV transpired in either intervention. Integrated substance use and IPV interventions, in contrast, showed reductions in IPV. However, one integrated intervention (Satyanarayana *et al.*, 2016) did not observe the same reductions in alcohol use. This latter finding is surprising given that experts have concurred that there are many implementation benefits to dually addressing IPV along with comorbid alcohol misuse that include streamlined services for patients, prevention of relapse, and benefits for families and children (Mootz *et al.*, 2022). While the findings of this review suggest that integrated substance use and IPV interventions show trends of also reducing IPV, more effectiveness research is needed.

Targeted and integrated mental health interventions for the most part showed improvements in mental health outcomes. About half of these interventions also improved IPV. Most mental health studies focused on reduction and measurement of depression symptoms as a primary outcome of interest, although only one study emphasized meeting criteria for depression as being necessary for eligibility to participate in the study (Patel *et al.*, 2017; Weobong *et al.*, 2017). It was more common to explicitly include alcohol use, HIV status, or IPV as eligibility criteria and assess depression within those specialized populations. We suggest that future studies not only include their primary outcomes of interest as eligibility criteria but also consider expanding measurement and the focus of treatment to be transdiagnostic and include all common mental conditions (depression, anxiety, and PTSD). These conditions are frequently comorbid (Yatham *et al.*, 2018) and all have strong bidirectional associations with family violence (Trevillion *et al.*, 2012).

Moreover, studies tended to emphasize intervention effects with less systematic evaluation of implementation factors, such as scalability and sustainability. Several studies, however, implemented task-shifting, having nonspecialized providers deliver mental health care, in settings outside the traditional mental health outpatient clinic. Task-shifting can improve scale-up efforts through nonspecialized delivery of care (Sijbrandij *et al.*, 2017). It can be challenging, however, to establish a proof of concept and simultaneously assess scalability and sustainability, especially for more complex, integrated interventions (Zamboni *et al.*, 2019). Going forward, hybrid implementation-effectiveness trials can evaluate both longitudinally given that a one-time scalability and sustainability assessment conducted at the beginning of a study is not recommended (Zamboni *et al.*, 2019). Continued consideration of how to adapt interventions to be relevant for co-occurring mental health conditions and family violence is recommended. An example of such an approach is the Common Elements Treatment Approach, a unified intervention designed for implementation in LMICs and based on evidence-based treatments for common mental conditions and substance use (Murray *et al.*, 2014).

Several limitations to this systematic review should be considered. While one of our main objectives was to identify effective interventions that addressed family violence and mental health

and/or substance use, our search strategy did not include other related problems, such as HIV prevention and response programming, that are related and relevant in low-income settings. While the intervention terms used should have identified most studies, it is possible that our inclusion criteria of pre- and post-measurement may have inadvertently occluded violence prevention studies that focused on implementation at the community level. Next, we only included studies with abstracts in English. Finally, although we identified studies from 13 LMICs, most of these studies were concentrated in sub-Saharan Africa.

## Conclusion

This systematic review of interventions that measured effects on family violence, mental health, and/or substance use included 19 studies (29 articles) located in 13 LMICs across three continents. Most of the studies were RCTs, were underpowered and did not blind to condition reducing the quality of the evidence. Most interventions focused on reduction of one problem (family violence, mental health, or substance use) and there was significant heterogeneity across studies. Most family violence studies focused on reducing IPV (rather than child maltreatment), and few interventions targeted IPV perpetration. Integrated substance use and IPV interventions had better success in reducing IPV than those that targeted substance use alone. About half of interventions that involved a mental health focus (mostly targeting depression) improved IPV. Future work should develop understanding of theoretical mechanisms that explain connections among these problems so that interventions can be adapted for a more comprehensive effect.

**Open peer review.** To view the open peer review materials for this article, please visit http://doi.org/10.1017/gmh.2023.62.

**Supplementary material.** The supplementary material for this article can be found at http://doi.org/10.1017/gmh.2023.62.

**Data availability statement.** The data that support the findings of this study are available on request from the corresponding author, J.M.

**Author contribution.** J.M.—Conceptualization, data extraction, review of articles, original draft writing; M.F.—data extraction, table construction, original draft writing; A.G.—original draft writing, risk of bias assessment; A.M.—literature search, original draft writing; C.G.—original draft writing, risk of bias assessment; M.W.—reviewing and editing.

**Financial support.** This project was supported by the National Institute of Mental Health K23 MH122661 and T32 MH096724, Global Mental Health Research Fellowship: Interventions That Make a Difference.

**Competing interest.** The authors declare no competing interests exist.

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
