## [Reviewer Report]

March 15, 2023

Dear Drs. Belkin, Bass, and Chibanda:

With pleasure, we submit our manuscript entitled “Interventions Addressing Family Violence and Mental Illness or Substance Use in Low- and Middle-Income Countries: A Systematic Review” for consideration for publication in Global Mental Health.

Family violence has strong associations with substance use and mental illness. Yet, interventions that comprehensively address these related problems are lacking. Most family violence research has been conducted in high-income countries, although family violence rates are higher in low- and middle-income countries (LMICs) and outcomes more severe. To determine effective treatments in LMICs, we conducted a systematic review of interventions that addressed family violence and mental illness or substance use in LMICs, identifying 29 articles produced from 19 studies conducted in 13 LMICs. We found that none of the studies showed improvements in all intervention areas. Child maltreatment was less likely to be addressed than intimate partner violence (IPV). Targeted interventions for substance and mental health mostly improved primary outcomes, although they were less effective in reducing IPV. While advances have been made, this review shows that much work is needed to develop and rigorously evaluate evidence-based treatments before innovations in implementation or scale-up efforts for family violence interventions and related mental health and substance use problems can occur.

This material has not been published elsewhere and is not under review at another publication.

Sincerely,

Jennifer Mootz and Co-Authors

---

## [Reviewer Report]

Overall, the authors are to be commended for a comprehensive review in an emerging area of intervention research. A few comments are offered to strengthen the manuscript.

1. The tables are difficult to comprehend and where possible, abbreviations should be removed replaced with full words. Alternative ideas would be for the first table on quality assessment to have green (yes), red (no), gray (unsure), yellow (partial) cell boxes rather than letters to quickly give the reader a sense of the strength of evidence. For the synthesis table, the target column is especially unclear and words should be used instead of letters and symbols as a reader should be able to understand the table quickly without needing to search for the legends. Additionally, the type column seems that primary and secondary (as full words) could be added to the outcome column as a descriptor and that column removed - this would allow for more space to be given to the target column. I would recommend the authors look at other systematic reviews for structure examples for this table.

2. The authors include allocation concealment and blinding participants and outcome assessment as part of the quality assessment which is fine. However, in practice, for these comprehensive, largely behavioral interventions, achieving this is very difficult. The authors should comment on this in the discussion rather than only implying studies are weak because of this. Additionally, a color coded table would more clearly indicate this is where all studies seem to have challenges at a quick glance by a reader.

3. The authors note a bidirectional relationship between mental health and family violence. There is limited mention of the importance of multiple time points of intervention follow-up for path analyses to unpack these relationships. Did any papers do this? What are the authors recommendations for this? If helpful, the authors should consider the inclusion of a diagram that illustrates multiple pathways between family violence and mental health.

4. Regarding mental health outcomes, could the authors add more in the discussion about whether these interventions prevent poor mental health or rather they ‘treat’ mental health symptoms or people already with specific diagnoses in the inclusion criteria of the interventions included? This would have implications for generalizability of the target population. For example, it is unclear if ‘most studies focused on reduction and measurement of depression symptoms’ means that people had depression or if these symptoms were just assessed in a general population.

5. For substance use, it is not clear that any intervention addressed substances outside of alcohol. If this is correct, the authors should describe the interventions as alcohol interventions rather than a more holistic substance use intervention.

6. The authors note limited studies on child maltreatment - what are the takeaways for children’s mental health? Do studies also measure this or only among adults? More analysis of this and recommendations in the discussion is warranted.

7. Please comment on the scalability, sustainability, and funding of these combined interventions in the analyses for stronger policy and practitioner recommendations.

---

## [Reviewer Report]

Thank you for this article, it is very well presented, and addresses a very important topic. I believe the authors should be commended for a good review, I have a few small comments noted below, and I believe addressing these will make for an excellent article worthy of publication.

My only content suggestion is the following: have you considered looking a the duration of interventions (both length of sessions and number of sessions) and effectiveness? Some research shows that very short interventions are often ineffective, and there may be a minimum amount of time that is needed for an intervention to be effective. I do not think it is essential to include this, but it did strike me that there would probably be an interesting comment to be made about this.

Consistency in language and terms – towards the beginning the authors use family violence, interpersonal violence, domestic violence – and they seem to be used inter-changeably. If you mean different things with the terms please define the difference or be consistent.

The authors usually use “mental health disorders” however occasionally you use “mental disorders”, which I believe can be perceived as derogatory language, I would suggest always using “Mental health disorders”

Introduction: “This is due in part to poverty which is pervasive in LMICs” Poverty is definitely one driver, but so is patriarchy or gendered norms – I think you should bring that in too, both the Jewkes and Gibbs papers discuss this aspect.

“Childhood abuse survivors or children who have witnessed IPV have shown to be more likely to perpetrate IPV themselves, placing their offspring at higher risk” they are also more likely to experience IPV, please add this.

Table 2 – I have table 1, but I do not see Table 2 – did I miss it or was it not uploaded?

Results:

“Both analyses found significant reductions of caregiver exposure to and conditions did not exist for parenting, family connectedness, conduct, or functional impairment” are there some words missing here?

---

## [Reviewer Report]

Thank you for addressing the comments. The tables in particular, are much easier to digest at first glance. I look forward to citing this paper in the future.

---

## [Reviewer Report]

Thank you for the comprehensive responses to the reviewers' comments. Your publication is now acceptable for publication subject to some minor editorial work. Please consistently use “mental health conditions” instead of “mental illness” and “mental health disorders” etc. Please review the following sentence in the discussion so that it makes sense “ Future interventionists and researchers should examine downstream effects of reducing IPV and adult mental health and substance use on children”